Alien species revises systematic status: integrative species delimitation of two similar taxa of Symbrenthia Hübner, [1819] (Lepidoptera, Nymphalidae)

Hsu Yu-Feng t43018@ntnu.edu.tw 1
Shen Zong-Yu 1 2 3
Huang Hang-Chi 4
Huang Chih-Wei 1
Lu Chen-Chih 5
1 Department of Life Science, National Taiwan Normal University , Taipei , Taiwan, R.O.C.
2 Biodiversity Research Center, Academia Sinica , Taipei , Taiwan, R.O.C.
3 Biodiversity Program, Taiwan International Graduate Program, Academia Sinica and National Taiwan Normal University , Taipei , Taiwan, R.O.C.
4 Butterfly Conservation Society of Taiwan , Taipei , Taiwan, R.O.C.
5 Fu-Xing Elementary School , New Taipei City , Taiwan, R.O.C.
Mitchell Andrew
Electronic publication date: 2023 Jan 30
Publication date: 2023
Volume: 11
Electronic Location ID: e14644
Received 2022 Sep 14; Accepted 2022 Dec 6
Copyright: ©2023 Hsu et al.
Copyright year: 2023
Copyright holder: Hsu et al.
License: This is an open access article distributed under the terms of the Creative Commons Attribution License, which permits unrestricted use, distribution, reproduction and adaptation in any medium and for any purpose provided that it is properly attributed. For attribution, the original author(s), title, publication source (PeerJ) and either DOI or URL of the article must be cited.
License URL: https://creativecommons.org/licenses/by/4.0/

Keywords: Systematics, Species delimitation, Biological introduction, Species interaction, Niche modeling

Funding: Council of Agriculture (Taiwan) 109-FM-08.2-C-16(1) Yangminshan National Park Headquarters 1100707 Taroko National Park Headquarters 1079007 The TIGP Biodiversity Program at Academia Sinica, Taiwan This project was supported by grants from the Council of Agriculture (Taiwan) (109-FM-08.2-C-16(1)), Yangminshan National Park Headquarters (1100707), Taroko National Park Headquarters (1079007), and the TIGP Biodiversity Program at Academia Sinica, Taiwan. The funders had no role in study design, data collection and analysis, decision to publish, or preparation of the manuscript.

==============================
Introduction of organisms to new range may impose detrimental effects on local organisms, especially when closely related species are involved. Species delimitation employing an integrative taxonomy approach may provide a quick assessment for the species status between taxa of interest, and to infer ecological competition and/or introgression that may be associated with the introduction. A nymphalid butterfly, Symbrenthia lilaea lunica, was recently introduced to Taiwan, where a closely related local taxon, S. l. formosanus, can be found. We employed multiple species delimitation methods to study the species status between the two taxa, and the results revealed that they can be recognized as two distinct species, revised to S. l. lilaea (syn. nov.) and S. formosanus (stat. rev.) respectively. We further performed a niche modeling approach to investigate the ecological interaction between the two species. The taxonomic status of the two taxa, now elevated to species, has been revised and conservation facing rapid expansion of the introduced species discussed.

Introduction

Biological invasions and range expansion of organisms usually impose unfavorable effects on local organisms that share similar ecological requirements (Mooney & Cleland, 2001), particularly when the expanding form is of continental origin and entering insular areas (Sax & Gaines, 2008). Competition may occur between native taxa and the invading one, notably when the counter taxa are closely related, where they are expected to share resource requirements (Zwerschke et al., 2018). This kind of scenarios have been well-documented in various organisms, such as ants (von Aesch & Cherix, 2005), birds (Koenig, 2003), molluscs (Zwerschke et al., 2018), plants (Leger & Espeland, 2010; Čuda et al., 2015; Sheppard & Brendel, 2021), etc., and even inspired E. O. Wilson to coin in his “taxon cycles” hypothesis (Wilson, 1961). Alternatively, if the involved taxa were previously allopatric populations or subspecies, gene flow may occur between taxa, resulting in changes in the genetic make-up of the local population (Rhymer & Simberloff, 1996). Consequently, whether the exotic taxon represents interspecific or intraspecific entity to the local taxon may lead to different types of ecological and evolutionary impacts to local community. Species delimitation employing an integrative taxonomy approach has becoming popular to help with taxonomic treatments/decisions among closely related taxa and to detect cryptic species (Moraes et al., 2021). This approach can also help with assessing whether an invading taxon is conspecific or not to the local taxon.

A case of recent invasion, either by anthropogenic introduction or by natural means such as wind, of a nymphalid butterfly currently recognized as Symbrenthia lilaea lunica Bascombe, Johnston & Bascombe, 1999 (Figs. 1A–1D) into Taiwan has been reported (Lu & Chen, 2014; Hsu et al., 2022). In Taiwan, there is a local native Symbrenthia Hübner, [1819] taxon, S. l. formosanus Fruhstorfer, 1908 (Figs. 1E–1H; Shirôzu & Ueda, 1992; Hsu et al., 2022). The taxon lunica (Bascombe, Johnston & Bascombe, 1999) is a replacement name for Papilio lucina Stoll, 1780; in Cramer, De uitlandsche kapellen 4(26b-28): 82 (Bascombe, Johnston & Bascombe, 1999), distributed from northern Indochina to southern China according to Tsukada & Nishiyama (1985). Although some authors (e.g., Bozano & Floriani, 2012; Ek-Amnuay, 2012; Lang, 2012; Monastyrskii, 2019) regarded lucina a synonym of the nominotypical lilaea described from India but have not formally synonymized these two names, others treat them as separate subspecies (e.g., Tsukada & Nishiyama, 1985; Hsu et al., 2022; Fric et al., 2022). Symbrenthia lilaea lunica from continental Asia was first documented in 2004 in Taiwan (Lu & Chen, 2014). Natural observations in Taiwan on immature biology and hostplant usage of both S. l. lunica and S. l. formosanus have documented that the two taxa use the same larval hostplants (Lu & Chen, 2014). The invading or introduction event represents a ‘natural experiment’ in which to examine whether these two taxa are conspecific as current classification suggests, or if they represent distinct biological species. The answer to this natural experiment may help clarify what ecological effect the introduced taxon may impose on the native taxa. If they turn out to represent distinct species, competition between them may occur as they do not differ in larval hostplant usage. If they turn out to be conspecific, gene introgression may occur since hybridization is liable to occur.

Figure 1 Adult specimens of Symbrenthia lilaea lunica and S. l. formosanus.

(A–B) Male specimen of S. l. lunica, TAIWAN: Miaoli County, Sanwan, Emei Bridge. (C–D) Female specimen of S. l. lunica, TAIWAN: Xinzhu County, Zhudong, 100 m. (E–F) Male specimen of S. l, formosanus, TAIWAN: Xinbei City, Wulai, 150 m. (G–H) Female specimen of S. l. formosanus, TAIWAN: Taoyuan City, Fuxing, Daman. Scale bar = 1 cm (A–H).

We investigated the species status between the two now sympatric taxa in Taiwan using multiple coalescent model based species delimitation methods and distance based species delineation, under a scheme for testing hypothesis of taxonomic status developed by Braby, Eastwood & Murray (2012), following by a niche modeling survey to understand the ecological interaction between two taxa.

Material and Methods

Sampling

The mitochondrial cytochrome oxidase subunit I (COI) gene has been successfully applied as a helpful marker with which to delimit closely related species (Hebert et al., 2003). We included a total of 13 specimens collected from various localities in our COI-based study. Four specimens of S. l. formosanus were collected around Taiwan, and six specimens of S. l. lunica were collected in Taiwan (mainland and Mazu archipelago), China, and Thailand. All samples were preserved in 70% ethanol and kept at −20 °C for the subsequent molecular study. Moreover, additional COI sequences of S. l. formosanus (AY788679) from Taiwan, S. l. lunica (EU368155, KJ649017, KX300094) from China, Vietnam, Myanmar respectively, and the nominate subspecies S. l. lilaea (Hewitson, 1864) (KP644228, KP644229) from India were obtained from GenBank. For the phylogenetic analyses, we used one sequence of S. brabira Moore, 1872 (EU368154) as an outgroup, which was also obtained from GenBank.

DNA extraction, PCR amplification and DNA sequencing

Genomic DNA was extracted from one leg of specimens using the Gentra Puregen tissue kit form QIAGEN (QIAGEN, Germantown, MD, USA), following the manufacturer’s protocol. A partial fragment from the COI gene was targeted for amplification by polymerase chain reaction (PCR). The COI gene was amplified using the universal primers COX-J-1460 (5′-TACAATTTATCGCCTAAACTTCAGCC-3′) and COX-N-2191 (5′-CCCGGTAAAATTAAAATATAAACTTC-3′). PCR reactions were performed in a 30 µL volume eppendorf, containing 1 µL of extracted DNA, 23.5 µL of ddH2O, 3 µL of 10X PCR reaction buffer, 0.6 µL of each primer and 0.3 µL of Power Taq (Genomics Biosci & Tech, Taiwan). The following PCR protocol was used: an initial denaturation at 95 °C for 5 min, followed by 40 cycles of 30s denaturation at 95 °C, 30s annealing at 50 °C and 45s extension at 72 °C, and a final extension at 72 °C for 10 min. Automatic sequencing was preformed using an ABI 3730XL DNA Analyzer (Applied Biosystems, Waltham, MA, USA).

Sequence analyses and phylogenetic reconstruction

Sequences were edited and assembled using Sequencher 4.10.1 (Gene Codes Corporation, Ann Arbor, USA), and sequence alignments were performed using MUSCLE in MEGA 11 (Tamura, Stecher & Kumar, 2021), and pairwise genetic distances between different populations of S. l. formosanus and S. l. lunica were also measured using MEGA 11 with the Kimura 2-parameter model.

The best-fit nucleotide substitution model for phylogenetic analysis was inferred using jModelTest 2.1.10 (Posada, 2008) based on Akaike information criterion (AIC). Phylogenetic trees were reconstructed under maximum likelihood (ML) and Bayesian inference (BI). ML analysis was performed using RAxML v8.2.10 (Stamatakis, 2014) with 1000 bootstrap replicates to assess the reliability of the tree. BI analysis was performed using MrBayes 3.2.6 (Ronquist et al., 2012). For MrBayes, the substitution model inferred from jModelTest was applied. The Bayesian Markov Chain Monte Carlo (MCMC) analysis for 109 generations with sampling every 1,000 generations was run to ensure the average standard deviation of split frequencies were below 0.01. The first 30% of trees were discarded as burn-in. FigTree v1.4.4 was used to visualize the consensus tree.

Molecular species delimitation analyses

Many molecular species delimitation programs have been proposed and broadly applied in speciation studies, which provides important evidence for integrative taxonomy. Among molecular species delimitation programs, the Poisson Tree Processes model (PTP) (Zhang et al., 2013), the Automatic Barcode Gap Discovery (ABGD) (Puillandre et al., 2012), and the Generalized Mixed Yule Coalescent model (GMYC) (Fujisawa & Barraclough, 2013) were developed as single locus-based approaches for species delimitation. Therefore, we delineated species limits among S. l. formosanus, S. l. lunica, and S. l. lilaea by employing the Molecular Operational Taxonomic Unit concept set by these three programs.

For PTP, we used the tree inferred by MrBayes as input tree on the web server (https://species.h-its.org/ptp/), with 100000 MCMC generations and 100 thining. Subsequently, PhyloMap was used to visualize the results of PTP. For ABGD, we performed the analyses on the web version of ABGD (https://bioinfo.mnhn.fr/abi/public/abgd/), with default settings of relative gap width (X = 1.5) and the Kimura two-parameter (K2P) model for nucleotide substitution. For GMYC, we used the phylogenetic tree inferred by MrBayes 3.2.6. The results from MrBayes were forced bifurcated by the “multi2di” and “chronos” function in the package “ape” in R 4.1.2. A single-threshold GMYC analysis was performed in the R package splits v1.0-20. We chose the single-threshold model because of the limited improvements of multiple-threshold model.

Species distribution model of S. l. formosanus and S. l. lunica in Taiwan

Symbrenthia lilaea lunica was not known to occur in Taiwan until recently, although it inhabits Mazu islands, which are small outlying islands of Taiwan and close to mainland Asia. However, S. l. lunica arrived to the main island of Taiwan due to anthropogenic activities or via natural dispersal, with the first credible record found in Xinzhu in northwestern Taiwan in 2004 (Lu & Chen, 2014). Since then, the range of S. l. lunica has expended quickly, and is currently found in lowland areas throughout Taiwan (Lu & Chen, 2014; Hsu et al., 2022). It is an interesting issue whether competitive exclusion has happened between S. l. lunica and native S. l. formosanus, especially if the species delimitation analyses decide they represent different species.

The occurrence data of S. l. formosanus and S. l. lunica were obtained from the Global Biodiversity Information Facility (GBIF) (https://gbif.org/, accessed 26 July 2021), Taiwan Moth Information Center (https://twmoth.tesri.gov.tw/peo/FBMothQueryP, accessed 26 July 2021), and the specimen collection at National Taiwan Normal University. To test the interaction between these two species, we separated the occurrence data into two stages based on year. Because the first documentation of S. l. lunica was in 2004, we divided the time period based on the median year (2012). The early invasion was defined as the data recorded from 1911 to 2012, and the late invasion was defined as the data recorded from 2013 to 2021. Repeated data was excluded using R 4.1.2, and we ensured the presences of only one presenting point in each raster to avoid overfitting. In total, 48 and 43 localities were obtained during the early invasion stage for S. l. formosanus and S. l. lunica, respectively (Fig. 2A), and 132 and 192 localities were obtained during the later invasion stage for S. l. formosanus and S. l. lunica, respectively (Fig. 2B). These data were organized using Microsoft Excel for the subsequent analyses.

Figure 2 Distribution points of Symbrenthia lilaea formosanus and S. lilaea lunica during two different invasive stages in Taiwan.

(A) The distribution points of S. l. formosanus and S. l. lunica in the early invasive stage (1911–2012). (B) The distribution points of S. l. formosanus and S. l. lunica in the late invasive stage (2013–2021). This figure was plotted following the methods in Lin (2018).

A total 19 bioclimate variables (period: 1979–2013) were collected from CHELSA (https://chelsa-climate.org/, accessed on 14 July 2021) at a spatial resolution of 30 arc-seconds (1 km2). These bioclimate variables were derived from temperature and precipitation, which are considered to be related to the distribution and survival of small arthropods and have been widely used in the prediction of species distribution (De Meyer et al., 2010; Xu et al., 2020). In order to avoid the effect of multicollinearity, these 19 variables were selected by the “vifstep” and “vifcor” function with the threshold of 10 and 0.6 separately in “usdm” package in R 4.1.2 (R Core Team, 2021; selected variables shown in Fig. 3).

Figure 3 The Jackknife of regularized training gain of different bioclimatic factors of Symbrenthia formosanus and S. lilaea between different invasive time stages.

(A) The jackknife results of S. formosanus in the early invasive stage. (B) The jackknife results of S. lilaea (=lunica) in the early invasive stage. (C) The jackknife results of S. formosanus in the late invasive stage. (D) The jackknife results of S. lilaea (=lunica) in the late invasive stage.

MaxEnt (3.4.4) (Phillips, Anderson & Schapire, 2006) was applied to predict the habitat suitability of S. l. formosanus and S. l. lunica based on the occurrence data. 10% of the data were selected to run a random test and the remaining data were run following the default settings. Presence-only data were generated pseudo-absences and 10,000 random background points were randomly selected by the MaxEnt model. The results were output after 10 cross-validation replicates.

The predictions generated from MaxEnt modeling were evaluated according to the threshold independent area under the receiver operating characteristic (ROC) curve (AUC) values. ROC curves were used to plot the true-positive rate against the false-positive rate and the AUC was used as a measure of the goodness of fit of the model. The AUC value ranges from 0 to 1, with higher values indicating higher predictive performance. The logistic output was chosen as an estimate of the probability of presence conditioned by bio-environmental variables per grid cell. Jackknifing was used to screen for the contribution of each bio-environmental variable used in the model.

We performed principal component analyses (PCA) to test the niche overlap of these two species in both the early invasion stage and late invasion stage. The 19 bioclimatic variables were obtained from the CHELSA database based on the GPS of each observation point. The analyses were conducted in R 4.1.2 using the function “prcomp”, with scatterplots built using the function “ggbiplot”. Additionally, in order to evaluate the niche shift pattern between the two Symbrenthia species in Taiwan, we apply methods modified from Bates, Ollier & Bertelsmeier (2020) to quantify the niche shift between S. l. formosanus and S. l. lunica by calculating niche overlap, presented by Schoener’s D, and niche expansion of S. l. lunica.

Results

Taxonomic decisions

Phylogenetic reconstruction of Symbrenthia COI samples (Fig. 4) revealed that all samples of lunica +lilaea form a monophyletic group sister to formosanus samples, which also formed a monophyletic group. The p-distance was 0.0017 between lunica and lilaea and 0.0505–0.0525 between lunica +lilaea and formosanus. PTP, ABGD and GMYC all recognize a two species scenario, with lunica +lilaea and formosanus each representing a distinct species. Therefore, formosanus is recognized as a species distinct from lunica +lilaea, with the combination as Symbrenthia formosanus Fruhstorfer, 1908 (stat. rev.). The taxon S. l. lunica (Bascombe, Johnston & Bascombe, 1999) is proposed to be lumped with S. l. lilaea, Hewitson, 1864 (syn. nov.) herein as the two may not be distinguished by COI barcode nor adult and immature morphology. We thus will call them S. formosanus and S. lilaea respectively in the remaining text of this article.

Figure 4 Systematic and species delimitation results of Symbrenthia lilaea and S. formosanus.

(A) Phylogenetic tree reconstructed by BI analysis of 20 COI sequences of Symbrenthia. Nodal support from the BI posterior probabilities is along with ML bootstrap pseudo-replicates (BI posterior probabilities/MLbootstrap). Columns after taxon names correspond to the results of single-locus species delimitation analyses: Poisson Tree Processes model (PTP), Automatic Barcode Gap Discovery (ABGD) and Generalized Mixed Yule Coalecent model (GMYC). (B) PhyloMap visualization the result of PTP species delimitation, including 20 taxa of Symbrenthia. The first axis (horizontal) explained 62.64% and the second axis (vertical) explained 36.08% of sequence variance among samples.

Environmental factors which contribute to the distribution of Symbrenthia species in question

After applying “usdm” package to remove those highly correlated factors, “bio 2”, “bio 8”, “bio 12” (Figs. 3A & 3B) are used to construct the species distribution model of “early invasive stage”, while “bio 7”, “bio 8”, “bio 9”, “bio 12”, “bio 18” (Figs. 3C & 3D) are used to construct the species distribution model of “late invasive stage”. According to the results of the jackknife test, the factors show different contribution patterns in the early invasive stage. In the early invasive stage, “bio 2” (annual precipitation) and “bio 12” (air temperature) contribute reversely between these two species; annual precipitation contributes more than mean diurnal air temperature range in the distribution model of S. formosanus, whereas mean diurnal air temperature range contributes more than annual precipitation amount in the model of S. lilaea.

Comparing the jackknife results of both species between the two invasive stages, “bio 8” contributes the most among all models. According to this, the mean daily air temperatures of the wettest quarter may play a key role in the distribution of these two Symbrenthia species in Taiwan.

The species distribution model and niche shifting of the two Symbrenthia species in different time stages

According to the species distribution model results, S. formosanus does not show an obvious change between the early and late invasive stages (Figs. 5A & 5C). For both invasive stages, the presence probability of S. formosanus seems to be higher in the suburban areas and places with less human activity. For S. lilaea, the distribution model presents different results between the two time stages (Figs. 5B & 5D). Particularly, presence probability in the southwest part of Taiwan is higher in the later invasive stage (Fig. 5D). The SDM results of both species show that the presence probability decreases in the Pingtung area, the southernmost county of Taiwan. Although there may be biological importance to this observation, it is most likely a result of uneven presence observation point density in the later stage. Most of the presence points for the late invasive stage SDM are from northern Taiwan.

Figure 5 The Species Distribution Model (SDM) of Symbrenthia formosanus and S. lilaea between different invasive time stages predicted from MaxEnt.

(A) The SDM of S. formosanus in the early invasive stage. (B) The SDM of S. lilaea in the early invasive stage. (C) The SDM of S. formosanus in the late invasive stage. (D) The SDM of S. lilaea in the late invasive stage.

From the results of the early and late invasive stages (Fig. 6), the niche overlap value increased during the recent years (past-2012 D: 0.48; 2013-2021 D: 0.64), and the niche expansion value of S. lilaea remained zero between the two different time stages. Together, these mean that, during these two periods of time, the niche of this alien species did not extend beyond the niche of the native species. According to the ENM model and the niche shift results, competitive exclusion seems to not be occurring between these two species over these 18 years.

Figure 6 Shifts in climatic niche between alien (Symbrenthia lilaea) and native (S. formosanus) species.

(A) The climatic niche shift between two Symbrenthia species in the early invasive stage. (B) The climatic niche shift between two Symbrenthia species in the late invasive stage. Red describes the climatic niche of S. lilaea and blue describes the climatic niche of S. formosanus.

Discussion

Taxonomic status of the introduced and native Symbrenthia butterflies

The introduced and native Symbrenthia butterflies in question of the study were regarded as conspecific subspecies prior to the present study (e. g., Hsu et al., 2022; Fric et al., 2022). It has been argued that species delimitation is difficult for allopatric populations or subspecies of similar forms (King, 1993; Braby, Eastwood & Murray, 2012), but in the present case, the introduction of continental S. lilaea to Taiwan has proven that insular S. formosanus ought to represent a species endemic to the island, instead of being a geographical race of the former. Distinctions between them include: (1) distal band on hindwing uppersides of both sexes form a continuous orange stripe in S. lilaea (Figs. 1A & 1C), whereas it is interrupted by darkened veins in S. formosanus (Figs. 1E & 1G); (2) distal tip of uncus is acute in S. lilaea (Fig. 7A), whereas it is blunt in S. formosanus (Fig. 7D); (3) distal margin of valva is rounded in S. lilaea (Fig. 7A), whereas it is angled, somewhat squared in S. formosanus (Fig. 7D); (4) ampulla is stout, slightly down-curved in S. lilaea (Fig. 7A), whereas it is slender, strongly bent downwards in S. formosanus (Fig. 7D); (5) posterior margin of sterigma is concave in S. lilaea (Fig. 7C), whereas it is truncate in S. formosanus (Fig. 7F); (6) yellow eggs are laid in cluster in S. lilaea (Fig. 8A), in contrast to green eggs laid singly in S. formosanus (Fig. 8D); (7) larvae are gregarious in S. lilaea (Figs. 8B–8C), but solitary in S. formosanus (Figs. 8E, 8F).

Figure 7 Genitalia of Symbrenthia lilaea and S. formosanus.

(A–B) Male genitalia of S. lilaea (Gen. Prep. JYL-877, NTNU). (C) Female genitalia of S. lilaea (Gen. Prep. JYL-880, NTNU). (D–E) Male genitalia of S. formosanus (Gen. Prep. YFH-1583, NTNU). (F) Female genitalia of S. formosanus (Gen. Prep. JYL-1036, NTNU). Scale bar = 1 mm (A–B; D–E), 1 mm (C; F).

Figure 8 Eggs and immature stages of Symbrenthia lilaea and S. formosanus.

(A) Egg mass of S lilaea. (B) Young larvae of S. lilaea. (C) mature larvae of S. lilaea. (D) Egg of S. formosanus. (E) Young larva of S. formosanus. (F) Mature larva of S. formosanus.

Vouchers

Symbrenthia formosanus Fruhstorfer, 1908	

Ssp. formosanus: XINBEI CITY [= TAIPEI Co.]: 1 ♀, Xindian, Sikanshui, 5. X. 2001 (H. S. Que leg.); 1 ♂, Wulai, 150 m, reared from Boehmeria nivea,emerged 8. XI. 2004, HSU 04L21 (Y. F. Hsu leg.); 1 ♂, Wulai, 13. II. 2005, reared from Debregeasia orientalis, emerged 12. III. 2005, HSU 05B6 (L. H. Wang leg.); 1 ♀, same locality, 5. VI. 2005, reared from B. densiflora,emerged 23. VI. 2005, HSU 05F9 (Y. F. Hsu leg.); 1 ♂, same locality, 5. VI. 2005, reared from D. orientalis, emerged 5. VI. 2005, HSU 05F10 (Y. F. Hsu leg.); 1 ♂, same locality, 13. VI. 2005, reared from B. nivea, emerged 2. VII. 2005, HSU 05F36 (Y. F. Hsu leg.); 1 ♀, same locality, 16. I. 2007 (Y. F. Hsu leg.); 1 ♂, Wulai, ca 200 m, 5. VI. 2005 (Y. F. Hsu leg.). TAIPEI CITY: 1 ♀, Daan, Baozangyan, 28. III. 2019 (L. Huang leg.). TAOYUAN CITY [= TAOYUAN Co.]: 1 ♂, Fuxing, Xuanyuan, ca 1000 m, 16. VI. 2005 (L. H. Wang leg.); 1 ♀, Fuxing, Daman, 10. VI. 2005, reared from B. nivea, emerged 22-23. VI. 2005, HSU 05F28 (Y. F. Hsu leg.); 1 ♀, same locality, 10. VI. 2005, reared from D. orientalis, emerged 23. VI. 2005, HSU 05F29 (Y. F. Hsu leg.); 1 ♀, Fuxing, Gaoyi, 600 m, 7. III. 2008 (Y. F. Hsu leg.). YILAN Co.: 1 ♂, 2 ♀, Nanao, 14Km S. Nanao, 6. VI. 2005, reared from B. nivea, emerged 8-17. VI. 2005, HSU 05F17 (Y. F. Hsu leg.); 3 ♂, 1 ♀, Datong, Qilan, 1. XII. 2005, reared from B. nivea, emerged III. 2006, HSU 05M7 (C. C. Lu & H. Y. Lee leg.). XINZHU Co.: 2 ♂, Jianshi, Xiuluan, 800 m, 21. VI. 2005 (L. H. Wang leg.); 1 ♀, Jianshi, Guanwu, Dalu Forest Trail, 2. X. 2006 (L. H. Wang leg.); 1 ♂, Jianshi, Yulao, ca 1500 m, 25. II. 2009 (C. K. Wang leg.). MIAOLI Co.: 1 ♂, Zhuolan, Liyutan, 300 m, 2. VII. 2006 (Y. F. Hsu leg.). TAIZHONG CITY [= TAIZHONG Co.]: 1 ♀, Heping, Shangguguan, 900 m, 30. X. 2006 (Y. F. Hsu leg.); 1 ♂, same locality, 29. VII. 2007 (Y. F. Hsu leg.); 1 ♂, Heping, Guguan, 700 m, 5. VII. 1998 (Y. T. Lo leg.); 1 ♂, Heping, Guguan, IV. 2005 (L. H. Wang leg.). NANTOU Co.: 1 ♂, Renai, Tunyuan/ Tianchi, 2000/2800 m, 31. VII- 1. VIII. 1998 (Y. T. Lo leg.); 1 ♂, 1 ♀, Renai, Meiyuan, ca 400 m, 22. XI. 2004, reared from B. nivea, emerged 16-22. XII. 2004, HSU 04L49 (Y. F. Hsu leg.); 1 ♀, Yuchi, Shiguanyin, 480 m, 22. XI. 2004, reared from B. densiflora, emerged 21. XII. 2004, HSU 04L52 (Y. F. Hsu leg.); 1 ♂, Lugu, Fenghuanggu, 600 m, 4. XII. 2004, reared from B. densiflora,emerged 23. XII. 2004, HSU 04M5 (Y. F. Hsu leg.); 1 ♂, 3 ♀, Lugu, Fenghuanggu, 4. XII. 2004, reared from B. nivea, emerged 17-20. XII. 2004, HSU 04M4 (Y. F. Hsu leg.), 1 ♀, same locality, 11. XII. 2004, reared from B. densiflora, emerged 14. II. 2005, HSU 04M14 (Y. F. Hsu leg.), 1 ♂, same locality, 12. XII. 2004 (Y. F. Hsu leg.); 1 ♂, Renai, Beidongyanshan, ca 1800 m, 17. XII. 2005 (L. W. Wu & Y. F. Hsu leg.) (genitalia preparation YFH 1583), 1 ♀, same locality, 17. XII. 2005, reared from D. orientalis, emerged 4. I. 2006, HSU 05M32 (Y. F. Hsu leg.); 3 ♂, Renai, Tunyuan, 1550 m, 28. VII. 2006 (Y. F. Hsu leg.); 1 ♀, Renai, Nanshanxi, ca 900 m, 1. IX. 2007 (Y. F. Hsu leg.); 1 ♂, Lugu, ca 600 m, 19. VII. 2006, reared from B. nivea, emerged 4. VIII. 2006, HSU 06G12 (Y. F. Hsu leg.); 1 ♂, 2 ♀, Lugu, ca 500 m, 1. I. 2007, reared from B, nivea, emerged 18. I. 2007, HSU 07A7 (Y. F. Hsu leg.); 1 ♀, Renai, Huisun, 600-700 m, 20. II. 2010 (Y. F. Hsu & H. C. Huang leg.). HUALIAN Co.: 2 ♂, Xiulin, Huitouwan, 23. VII. 2005 (Y. F. Hsu leg.), 1 ♂, same locality, 30. I. 2007 (Y. F. Hsu leg.); 4 ♂, 3 ♀, Xiulin, Zuocang Trail, 7. III. 2006 (Y.F. Hsu leg.); 1 ♀, Xiulin, Guangbeibabiao, 2200 m, 28. V. 2007 (Y. F. Hsu leg.); 1 ♀, Xiulin, Huoranting, 1000 m, 10. XII. 2007 (M. H. Sun leg.); 1 ♂, Xiulin, Wujiabengshan, 2000 m, 7. VI. 2008 (L.H. Wang leg.); 1 ♀, Zhouxi, Zhongping Forest Trail, 1. VI. 2007 (L. H. Wang leg.). TAINAN CITY [= TAINAN Co.]: 1 ♀, Dongshan, Kantoushan, 600-800 m, 4. II. 2010 (Y. F. Hsu leg.). PINGDONG Co.: 1 ♂, Shizi, Nunaishan, 200 m, 26. III. 2006 (Y. F. Hsu leg.); 1 ♂, Fangliao, Yuquan, 26. II. 2006 (Y. F. Hsu leg.); 2 ♂, 1 ♀, Shizi, Lilongshan, 500 m, 17. II. 2008 (Y. F. Hsu leg.); 1 ♂, same locality, 1. III. 2008 (Y. F. Hsu leg.); 1 ♀, Wutai, Jiudawu, 500 m, 11. V. 2008 (C. L. Huang leg.); 1 ♀, Sandimen, ca 300 m, 2. I. 2009 (Y. F. Hsu leg.). TAIDONG Co.: 1 ♀, Yanping, Hongye, 11. I. 2009 (C. H. Lin, jr leg.); 1 ♀, Taidong, Pipa Lake, 4. VI. 2006 (Y. F. Hsu leg.); 1 ♀, Haiduan, Liyuan, 7. VI. 2020 (Y. F. Hsu leg.) (genitalia preparation JYL1036).

Symbrenthia lilaea (Hewitson, 1864)

JILONG CITY: 1 ♀, Longgang Trai, 26. IX. 2006 (Y. F. Hsu leg.); 2 ♀, same locality, 2. X. 2006 (Y. F. Hsu leg.);1 ♀, same locality, 7. II. 2007 (C. K. Wang leg.); 1 ♂, Shenao, Sea Level, 20. VI. 2006, reared from B. nivea,emerged 2. VII. 2006, HSU 06F17 (Y. F. Hsu leg.); 2 ♂, Haimentianxian, 26. IX. 2006 (Y. F. Hsu leg.). XINBEI CITY [= TAIPEI Co.]: 7 ♂, 7 ♀, Wulai, 150 m, 8. XI. 2004, reared from B. nivea, emerged 29. XI-5. XII. 2004, HSU 04L20 (Y. F. Hsu leg.); 1 ♂, Wulai, 11. III. 2005 (C. R. Chen leg.), 1 ♀, same locality, 5. VI. 2005 (Y. F. Hsu leg.), 1 ♂, 1 ♀, same locality, 11. VIII. 2005, reared from B. densiflora,emerged 25. VIII. -1. IX. 2005, HSU 05H8 (Y. F. Hsu leg.), 1 ♂, same locality, 23. IX. 2005, reared from B. nivea, emerged 4. X. 2005, HSU 05J72 (J. R. Chen leg.), 3 ♀, same locality, 6. X. 2006, reared from B. nivea, emerged 3-15. XI. 2006, HSU 06L3 (Y. F. Hsu leg.); 1 ♂, Wulai, Fushan, 700 m, 4. VII. 2005, reared from B. nivea, emerged 15. VII. 2005 (C. L. Huang leg.); 1 ♀, Wulai, Fushan, 16. VI. 2005 (C. L. Huang leg.); 1 ♂, Wulai, Baoqing Temple, ca 620 m, 4. VIII. 2005 (J. R. Chen leg.); 1 ♀, Xindian, Yinhedong, 250 m, 14. VI. 2005 (Y. F. Hsu leg.); 8 ♂, 1 ♀, Pinglin, Yuguang, 2. XI. -11.XII 2005, reared from B. formosana, emerged 29. XI. /18. XII. 2005, HSU 05L5 (C. C. Lu leg.); 1 ♀, Pinglin, Zhongxinlun, 2. XI. 2005 (C. C. Lu leg.). TAIPEI CITY: 1 ♀, Beitou, Guizikeng, 24. X. 2004 (H. C. Huang & P. Lo leg.); 2 ♀, Wenshan, Xianjiyan, 16. V. 2005 (Y. F. Hsu leg.); 2 ♀, Daan, NTNU campus, 9. VI. 2005, reared from B. nivea, emerged 23-24. VI. 2005, HSU 05F18 (Y. F. Hsu leg.); 5 ♀, Wenshan, Gongguan campus, NTNU, 9. VI. 2005, reared from B. nivea, emerged 21-24. VI. 2005, HSU 05F18 (Y. F. Hsu leg.); 3 ♀, same locality, 15. I. 2007, reared from B. nivea, emerged 31. I. 2007, HSU 07A20 (Y. F. Hsu leg.); 1 ♂, Shilin, Tianxiyuan, 17. VII. 2011 (H. C. Huang leg.); 1 ♂, Neihu, Dagouxi, 25. I. 2014 (L. Huang leg.). TAOYUAN CITY [= TAOYUAN Co.]: 2 ♀, Fuxing, Kapu, 800 m, 14. X. 2007, reared from B. densiflora, emerged 29. XI. 2007, HSU 07K7 (Y. F. Hsu & H. C. Huang leg.). YILAN Co.: 1 ♂, Yuanshan, Fushan Botanical Garden, ca 700 m, 4-5. VIII. 2006 (Y. F. Hsu & H. C. Huang leg.); 1 ♂, 1 ♀, Toucheng, Guishan Is., 4-5. III. 2006 (H. C. Huang & C. L. Huang leg.); 3 ♀, same locality, 24. VI. 2006 (H. C. Huang leg.); 1 ♀, same locality, 24. VI. 2006 (C. L. Huang leg.); 2 ♀, same locality, 4. VIII. 2007 (H. C. Huang leg.). XINZHU Co.: 1 ♂, Qionglin, Feifengshan, ca 80 m, 19. I. 2005, reared from B. densiflora, emerged 14. II. 2005, HSU 05A4 (Y. F. Hsu leg.); 1 ♂, Qionglin, Feifengshan, 10-80 m, 19. I. 2005, reared from B. nivea, emerged 8. II. 2005, HSU 05A3 (Y. F. Hsu leg.); 1 ♀, Zhudong, 11. IV. 2006 (Y. F. Hsu leg.); 2 ♀, Zhudong, 100 m, 28. III. 2008 (Y. F. Hsu leg.). MIAOLI Co.: 8 ♂, 2 ♀, Nanzhuang, 200 m, 28. XI. 2004, reard from B. densiflora, emerged 11-20. XII. 2004, HSU 04L68 (Y. T. Lo leg.); 1 ♂, Touwu, Xiangshan Village, Jinshui, 60 m, 27. II. 2005, reared from B. nivea, emerged 7. III. 2005, HSU 05B36 (Y. F. Hsu leg.); 5 ♂, 4 ♀, Sanwan, Emei Bridge, 14. I. 2006, reared from B. nivea, emerged 3-7. II. 2006, HSU 06A12 (Y. F. Hsu leg.); 1 ♂, Zhuolan, Liyutan, 2. VII. 2006 (Y. F. Hsu leg.). NANTOU Co.: 2 ♂, 3 ♀, Lugu, 500 m, 1. I. 2007, reared from B. nivea, emerged 17-31. I. 2007, HSU 07A6 (Y. F. Hsu leg.). HUALIAN Co.: 1 ♀, Wanrong, Wanrong Forest Trail, 29. VIII. 2010 (Z. H. Yen leg.). TAINAN CITY [= TAINAN Co.]: 2 ♀, Xinhua, Dakeng, 80 m, 10/11. II. 2006 (Y. F. Hsu leg.); 1 ♀, Xinhua, 4. II. 2006, reared from B. nivea, emerged 13. III. 2006, HSU 06J41 (Y. F. Hsu leg.), 2 ♂, same locality, 20. I. 2007 (Y. F. Hsu leg.). KAOHSIUNG CITY: 3 ♂, 1 ♀, Jiaxian Dist., Butingshan, ca 700 m, 26. VII, 2022, reared from Pouzolzia elegans, emerged 14. VIII. 2022, HSU 22G36 (Y. F. Hsu leg.). LIANJIANG Co.: 1 ♂, 1 ♀, Nangan, 10. VII. 2003, reared from B. nivea, emerged 25. VII. 2003, HSU 03G20 (L. W. Wu leg.); 1 ♂, Nangan, Shengli Dam, 9-10. XII. 2006, reared from B. nivea, emerged 6. I. 2007, HSU 06M2 (H. C. Huang leg.); 1 ♂, 1 ♀, Beigan, Qinbi, 22. IV. 2007 (Y. F. Hsu & H. C. Huang leg.).

Niche overlap between two Symbrenthia species in Taiwan

Many studies have reported that when a newly introduced species is present, competitive exclusion could be observed between the alien and similar native species (Mooney & Cleland, 2001; Paini, Funderburk & Reitz, 2008; Muthukrishnan, Hansel-Welch & Larkin, 2018). However, our study presents a different aspect of this interaction. Both the SDM and the niche overlap results showed that the degree of overlap of the presenting area between these two species increases over time (Fig. 6). This means that competitive exclusion may not be present between the alien and the native species. This result may be explained by the following two alternative scenarios:

Firstly, obvious competition for the two species may not be observed due to insufficient time of introduction of the alien species. S. lilaea was first found on the main island of Taiwan as recently as 2004, and thus may still be in the process of population establishment and early growing stages (McGeoch & Jetz, 2019). Consequently, the competitive exclusion effect between these two species may have not yet occurred or not yet occurred to a level observable by our available data.

Secondly, perhaps no competitive exclusion will occur between the two Symbrenthia species due to abundant host plant resources. Some studies have shown that host plants are much more important for the distribution of herbivorous insects when compared with the abiotic environmental factors (Wiens et al., 2010; Simões & Peterson, 2018). These two butterfly species feed on several species in the family Urticaceae, and most of these host plants are common and abundant in Taiwan (Yang, Lin & Liu, 1996). The food supply to the caterpillars of Symbrenthia may therefore be beyond the demand of both species combined, resulting in the absence of interspecific competition.

It awaits to be seen which scenario is more likely to occur, but it may be worthwhile to notice that S. formosanus was abundant in the Yangmingshan National Park of northern Taiwan (Chang, 1994), but in a butterfly survey conducted with sampling on monthly basis there from the beginning of 2021 to date, only S. lilaea has been recorded (Hsu et al., unpublished data). This observation suggests competition between the two species may actually have initiated.

Distribution difference of S. lilaea between two different invasive stage

The SDM of S. lilaea shows different result patterns between the early and late invasive time stages (Figs. 5B & 5D). Especially in southwest portion of Taiwan, the presence probability increases significantly in the late invasive stages. This phenomenon can be the result of the expansion of the distribution area of this alien species. The first record of S. lilaea is in Xinzhu county located in the northwest part of Taiwan. The distribution area of this species gradually expanded during the 18 years since it was first observed in Taiwan, and this butterfly species can today be observed in nearly all lowland areas around Taiwan.

The species distribution model (SDM) has been widely used as a tool to detect the potential invasive area of invasive species (Wiens et al., 2010; Ahmed, Atzberger & Zewdie, 2020). Based on the niche conservatism of the invasive species, we are usually able to predict the invasive area based on areas in the invaded region with similar environment to the source area from which the species originated. However, the SDM difference of S. lilaea between two different invasive stages suggests that the SDM may have inaccurately predicted potential areas of invasive species presence. The niche may be hard to quantify, even though some studies have suggested methods to measure it (Fraimout & Monnet, 2018; Lei et al., 2019), but introduced species usually still undergo niche expansion in the newly invaded area (Datta, Schweiger & Kühn, 2019; Bates, Ollier & Bertelsmeier, 2020). Although SDM is still a widely used tool to evaluate potential impact of invasive species, the inaccuracy of the model is inevitable due to the reasons addressed above. We suggest to include more biotic factors of the invasive species when predicting potential invasive regions rather than relying on the SDM results alone. By combining the biotic variables with the SDM constructed by the abiotic variables, the results should be closer to the realistic distribution pattern.

Conclusions

Species delimitation employing an integrative taxonomy approach has helped to clarify taxonomic entities of an introduced and a native Symbrenthia butterfly taxa regarded conspecific to date, leading to a decision to recognize each as a distinct species. This result suggests that interspecific competition may occur by the introduction of the alien species, rather than gene introgression. Subsequently, a niche modeling investigation was following, and the result showed that competition between the two species interest has not yet occurred or just initiated.

Supplemental Information

Supplemental Information 1 The niche overlap results infer from the 19 bioclimatic variables by principal component analyses (PCA)

(A) The niche overlap results of the early invasive stage. (B) The niche overlap results of the late invasive stage.

Click here for additional data file.

Supplemental Information 2 The receiver operating characteristic curve (ROC) and the threshold independent area under the ROC curve (AUC) values of the SDM

(A) the ROC and AUC of S. formosanus in the early invasive stage. (B) the ROC and AUC of S. lilaea (=lunica) in the early invasive stage. (C) the ROC and AUC of S. formosanus in the late invasive stage. (D) the ROC and AUC of S. lilaea (=lunica) in the late invasive stage.

Click here for additional data file.

Supplemental Information 3 Pairwise distance of Symbrenthia samples

Click here for additional data file.

Supplemental Information 4 Symbrenthia samples information

Click here for additional data file.

Supplemental Information 5 Raw data and code for GMYC analysis

Click here for additional data file.

Supplemental Information 6 Raw data and analysis for Species Distribution Model (SDM)

Click here for additional data file.

We are grateful to Dr. Jen-Pan Huang, Mr. Ming-Hsun Chou (both from Academia Sinica) and Mr. Trevor Padgett (Academia Sinica & Tunghai University) for reading and improving the manuscript of this article. We express our gratitude to Mr. Min-Wei Chai (National Taiwan Normal University) for giving us lots of advice on SDM. We also thank Dr. Masaya Yago (The University Museum, University of Tokyo) for helping us with valuable literatures. Also, we express our gratitude for the DNA Sequencing Core Facility of the Institute of Biomedical Sciences of Academia Sinica, Taipei, for providing DNA sequencing service.

Additional Information and Declarations

Competing Interests

Author Contributions

DNA Deposition

Data Availability

The authors declare there are no competing interests.

Yu-Feng Hsu conceived and designed the experiments, performed the experiments, analyzed the data, prepared figures and/or tables, authored or reviewed drafts of the article, and approved the final draft.

Zong-Yu Shen conceived and designed the experiments, performed the experiments, analyzed the data, prepared figures and/or tables, authored or reviewed drafts of the article, and approved the final draft.

Hang-Chi Huang performed the experiments, prepared figures and/or tables, performed field work and prepared figures, and approved the final draft.

Chih-Wei Huang performed the experiments, prepared figures and/or tables, performed field work and compiled data, and approved the final draft.

Chen-Chih Lu performed the experiments, prepared figures and/or tables, performed field work and prepared figures, and approved the final draft.

The following information was supplied regarding the deposition of DNA sequences:

The COI sequences are available at GenBank: LC727399 to LC727411.

The following information was supplied regarding data availability:

The raw data and code for the analysis in this study are available in the Supplementary Files.

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
