# Peer review of "Alien species revises systematic status: integrative species delimitation of two similar taxa of Symbrenthia Hübner, [1819] (Lepidoptera, Nymphalidae)"

_PeerJ, doi:10.7717/peerj.14644_

## Round 0.1 · original submission · Minor Revisions

The reviewers have done a thorough job and I agree with their assessment. The manuscript addresses the main issue comprehensively, however the reviewers provide many useful suggestions for improvement. My only addition is that you should mention in the text (not only in the phylogenetic tree, Fig. 5A) what the GenBank numbers are of the sequences you derived for this study.

·

Basic reporting

Manuscript #77026 (“Alien species revises systematic status: integrative species delimitation of two similar taxa of Symbrenthia Hübner, [1819] (Lepidoptera, Nymphalidae)”) brings interesting results about the distribution and status of Symbrenthia lilaea in Taiwan. The authors documented that the Taiwanese population of the butterfly consists of individuals of different origins, one endemic and the other originating from the Asian continent. The authors showed that these represent rather different species, they elevated the endemic form formosanus to a species rank and synonymized the invasive S. l. lunica with a nominotypical species. This finding is supported by molecular investigation using classic barcoding, but also by such detailed information as the appearance of the adults and their genital structures, both male and female, and by documenting the differences in the life cycle. Furthermore, the authors revised the distribution of both taxa in Taiwan, calculated Species Distribution Model using the Maximum Entropy algorithm, and compared the distribution from two different time sets (before and after 2013).
The paper is well written and well organized. The results are very interesting, explaining the details of the two Taiwanese taxa. The findings fit well with the results by Fric et al. (2022) and explain the strange pattern from that paper, i.e., Taiwan hosts two different kinds of taxa from the Symbrenthia lilaea complex. There are only a few omissions – the authorities of both studied taxa should be mentioned already in the Introduction part, and the rationale for making two different sets of data for the SDM model (first and second invasive phase) should be more explained. There are numerous typos all over the text, which should be corrected.

Experimental design

no comment

Validity of the findings

no comment

Additional comments

no comment

Reviewer 2 ·

Basic reporting

Indeed, the paper is well written and well organized. The figures and tables are relevant, and comprehensive with good publishable resolution. The paper solved an old taxonomic ambiguity of the genus Symbrenthia, and also intends to show the ecological interaction between native and invasive species.

The introduction is reasonably good, however, the background history of the species S. l. lunica is missing from the introduction, as it is an introduced species to the island. The species S. l. lunica (Bascombe, Johnston & Bascombe, 1999) is a replacement name for Papilio lucina Stoll, 1780; in Cramer, Uitl. Kapellen 4(26b-28):82. Importantly, it is also noticeable that some authors already considered P. lucina as a synonym of S. lilaea. Here at this point, I would suggest the authors add a brief history note of the species following this question, why these earlier treatments are not accepted here, and what are the lacuna of those treatments?

One overarching issue with the citation and reference is that the paper doesn’t follow the journal rule properly, I suggest the author(s) should pay attention to it.

Finally, I would suggest the author(s) get some help from a colleague or professional editing service to improve the English language.

Experimental design

The research questions are relevant and well defined. In general, the experimental design is clearly written and perfectly fits with the research questions. However, some minor changes, additions, and modifications would be suggested as follows

Line 82. GenBank information on the sequence data should be clearly mentioned within the text.

Line 86. The author should provide the source of the sequence of the species S. brabira

Line 152: It is somewhat ambiguous that the years are randomly divided into two stages. Therefore, it needs clarification. Here, I suggest the authors should add a few words on these two stages; on what basis author(s) divided the years, and why is 2012 considered as a boundary???

The other observations and concerns are stated in the comments of the PDF.

Validity of the findings

The results of the paper are promising and defiantly addressed the research questions. The taxonomic treatments are well stated and the ecological experiments are interesting, and further study could be more interesting and useful for understanding the invasive species ecology.

Additional comments

Author(s) should give attention in applying punctuation marks not only in the reference section but for the whole text also. Many places’ author(s) misuse/misapplied “period” and “comma”.

When author mentioned a range, it should be indicated by “en dash”, not other symbol like “hyphen”, “em dash”. It seems that author(s) use “em dash” in few places, the author should be careful about this.

Line 43. The full reference of citation "Koenig, 2003" is missing, the author(s) need to check carefully and provide the full reference of the citation

Line 43. The citation "von Aesch & Cherix, 2005" doesn't match with provided reference. I suggest that author(s) should provide an appropriate citation in the text.

Line 44. Author(s) should follow a uniform pattern within the text, if "diacritic" is used in name(s) then it should reflect in both citation and reference as well. The author(s) need to check and correct it.

Line 55. While a genus or a species is introduced/mentioned for the first time in the text, it should be with author(s) and year of the particular genus or species. Here I suggest that the author(s) should mention the author and year of the species.

Line 268. Seems confusing: why slash??? is it for the conjunction "or", if so, I would suggest to use "en dash", and it should be same for the rest.

Line 556-558. Species name should be same in both Figure title and subtitle, and it should match with text section also e.g., in text, 157–158 line, author(s) follow trinomial nomenclature so it should be same in figure section as well. And it should be same for the rest of figures also.

Line 586-587. While the species name S. lilaea is mentioned within the text as well as in the figure title. So, it must be the same within the figures also. I suggest the author(s) change the “B” and “D” heading within the figure into “lilaea” or “lilaea (=lunica)”

For the other minor comments please follow the pdf with comments.

Annotated reviews are not available for download in order to protect the identity of reviewers who chose to remain anonymous.

---

## Round 0.2 · accepted · Accept

The authors have followed all of the reviewers' suggestions and the manuscript is now much improved and ready for publication.